# Electrochemical Aspects of a Nitrogen-Doped Pseudo-Graphitic Carbon Material: Resistance to Electrode Fouling by Air-Aging and Dopamine Electro-Oxidation

**Kailash Hamal [1], Jeremy May [1], Haoyu Zhu [2], Forrest Dalbec [1], Elena Echeverria [3], David N. McIlroy [3], Eric Aston [4] and I. Francis Cheng [1],\***

[1] Department of Chemistry, University of Idaho, 875 Perimeter Dr, MS 2343, Moscow, ID 83844, USA; hama7198@vandals.uidaho.edu (K.H.); may9876@vandals.uidaho.edu (J.M.); dalb6827@vandals.uidaho.edu (F.D.)

[2] Department of Material Science and Engineering, Boise State University, MEC 403D, 1910, Boise, ID 83725, USA; haoyuzhu@boisestate.edu

[3] Department of Physics, Oklahoma State University, 145 Physical Sciences Building, Stillwater, OK 74028, USA; elena.echeverria@okstate.edu (E.E.); dave.mcilroy@okstate.edu (D.N.M.)

[4] Department of Chemical and Materials Engineering, University of Idaho, 875 Perimeter Dr, MS 2343, Moscow, ID 83844, USA; aston@uidaho.edu

\* Correspondence: ifcheng@uidaho.edu; Tel.: +1-208-885-6387

**Abstract:** The nitrogen-doped form of GUITAR (pseudo-Graphite from the University of Idaho Thermalized Asphalt Reaction) was examined by X-ray photoelectron, Raman, and X-ray diffraction spectroscopies and cyclic voltammetry (CV). Electrochemical studies indicate that N-GUITAR exhibits significant resistance to fouling by adsorption and by passivation. Unlike other carbon materials, it maintains fast heterogenous electron transfer (HET) kinetics with $Fe(CN)_6^{3-/4-}$ with exposure to air. The CV peak potential separation ($\Delta Ep$) of 66 mV increased to 69 mV in 3 h vs. 67 to 221 mV for a highly oriented pyrolytic graphite (HOPG) electrode. Water contact angle measurements indicate that N-GUITAR was able to better maintain a hydrophilic state during the 3-h exposure, going from 55.8 to 70.4° while HOPG increased from 63.8 to 80.1°. This indicates that N-GUITAR better resisted adsorption of volatile organic compounds. CV studies of dopamine also indicate N-GUITAR is resistant to passivation. The $\Delta Ep$ for the dopamine/o-dopaminoquinone couple is 83 mV indicating fast HET rates. This is reflected in the peak current ratios for the oxidation and reduction processes of 1.3 indicating that o-dopaminoquinone is not lost to passivation processes. This ratio along with the minimal signal attenuation is the best reported in literature.

**Keywords:** adsorption; dopamine; electrode fouling; electrode kinetics; nitrogen doping; pseudo-graphite

## 1. Introduction

GUITAR (pseudo-Graphite from the University of Idaho Thermalized Asphalt Reaction) is a nanocrystalline graphite-like hydrogenated amorphous carbon (85% sp$^2$ and 15% sp$^3$ carbon) described in detail in recent publications [1–7]. It has morphological features similar to classical graphites but has divergent chemical and physical properties. Most prominent among these is that GUITAR has fast heterogeneous electron transfer (HET) rates at its basal plane. The defect-rich nature of GUITAR results in a high density of electronic states (DOS) near its Fermi-level. Graphitic materials have a low DOS or they are subject to air-aging, either effect creates sluggish HET kinetics [8–11]. Another feature

is the resistance to corrosion that surpasses graphitic materials by 3 orders of magnitude in corrosion currents, and surpasses the potential windows of graphitic materials by 1 V [1,2]. In these respects it equals sp$^3$ carbon electrodes. In summary, it has the best combination of fast HET and high corrosion resistance of any electrode [1]. Recent studies have indicated that introduction of nitrogen impurities within the carbon lattice significantly increases DOS thus improving HET rates [12–14]. This effect (not yet examined on GUITAR) has been used to improve electrochemical sensor performance, oxygen reduction reaction (ORR) catalysts in fuel cells, batteries, and ultracapacitors [12].

Electrode fouling is the process by which current signal attenuation occurs when sample matrix components undergo either passive adsorption or by redox processes which forms a polymeric film on the surface [11,15–18]. Both create a barrier to electron transfer. Recent studies indicate that fouling process of highly oriented pyrolytic graphite electrodes (HOPG) when exposed to laboratory air is through adsorption of volatile organic compounds (VOCs), which decrease HET rates with the $Fe(CN)_6^{3-/4-}$ redox probe [11,15]. The important biologic analyte, dopamine undergoes electron transfer that gives rise to a cascade of reactions that result in an insulating layer of polydopamine [18,19]. There is a constant search for electrodes that exhibit minimized fouling characteristics. Such materials will find eventual use in long-term sensors and implantable electrodes for in-vivo analyses and bionics [20–22]. For this investigation we synthesized, characterized and examine nitrogen-doped GUITAR (N-GUITAR) electrodes for its HET rates, resistance to corrosion and fouling by air-aging and with dopamine voltammetry.

## 2. Materials and Methods

### 2.1. Materials and Chemicals

The reagents consisted of $N_2$(g) (>99.5%, Oxarc, WA, USA), acetonitrile (Fisher Scientific, Fair Lawn, NJ, USA), dopamine hydrochloride (Sigma-Aldrich, St. Louis, MO, USA). Potassium monophosphate (99.6%), potassium diphosphate (99.8%), potassium chloride (99.7%) (Fisher Scientific, Waltham, NJ, USA) and potassium ferricyanide (Acros Organics, Morris Plains, NJ, USA). All were used as received. The quartz tube was obtained from Technical Glass products, Inc. (Painesville Township, OH, USA), cut into small wafers (~2 cm × 0.5 cm) and used as deposition substrate.

### 2.2. Synthesis of Nitrogen-Doped Pseudo-Graphite (N-GUITAR)

N-GUITAR samples were prepared via a chemical vapor deposition (CVD) method. This apparatus is shown in Figure 1. The target substrate was loaded into a quartz boat and the tube furnace was heated to 900 °C with $N_2$ gas for 10 min prior to addition of acetonitrile vapor. This system was sealed with ceramic wool to minimize air back flow. Deposition commenced by stopping the $N_2$ flow and allowing the passage of acetonitrile vapor from a round bottom flask at >80 °C for 25 min. The $N_2$ flow is resumed and tube furnace was cooled to room temperature before coated sample were removed.

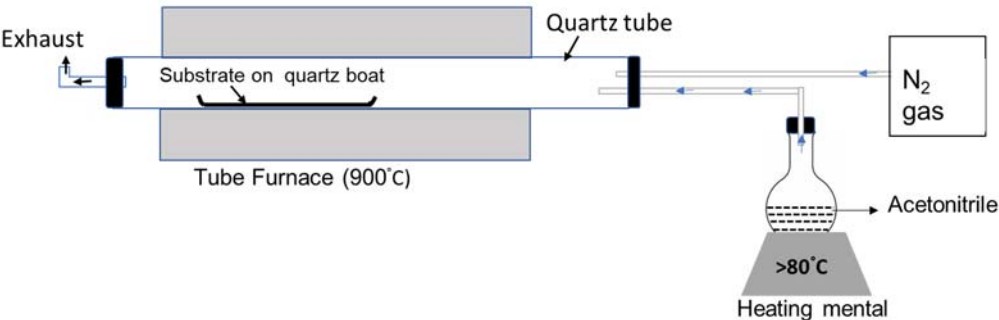

**Figure 1.** Schematic for synthesis of N-doped pseudo-graphite using a modified CVD technique.

### 2.3. Scanning Electron Microscopy (SEM)

These images were recorded with a FEI Teneo field emission microscope operating at an accelerating voltage of 2 kV, a spot size of 25–50 pA, with a secondary electron detector. Peeled HOPG thin sheet, GUITAR and N-GUITAR flakes were attached to a mounting disk with double-sided carbon tape.

### 2.4. X-ray Photoelectron Spectroscopy (XPS)

Spectra were recorded at Oklahoma State University in a custom-built vacuum chamber with a base pressure of $5 \times 10^{-9}$ torr. Measurements were acquired with the Al K$\alpha$ emission line (1486.6 eV) and a hemispherical energy analyzer with a resolution of 25 meV. Spectra were acquired at room temperature. Deconvolved C1s and N1s XPS peaks were fitted to gaussian curves with Shirley background subtraction and with the full width at half maximum kept constant.

### 2.5. Raman

The Raman spectra was collected using a WITec™ alpha300 R Raman instrument (Ulm, Germany), UHT-300 spectrometer with an Andor™ DU970N-BV CCD detector. The excitation source was a 532.5 nm, 100-mW frequency-doubled Nd:YAG laser.

### 2.6. Powder X-ray Diffraction (XRD)

XRD was conducted on a Siemens D5000 Diffractometer (Germany). The spectra were taken with Cu K-alpha radiation (0.154 nm) at 40 kV and 30 mA in the range of 2$\theta$ = 2–80° and step size of 0.05° at room temperature.

### 2.7. Water Contact Angle Measurement (WCA)

Contact angle measurements were conducted with a PG-2 Pocket Goniometer (Thwing-Albert, West Berlin, NJ, USA) under ambient conditions in dynamic mode using the sessile water drop (4–9 µL) method on the carbon surfaces. Contact angle was measured by capturing images over a 6 s period using Pocket Goniometer v.3.3 software (Thwing-Albert, West Berlin, NJ, USA) [23]. The GUITAR and N-GUITAR samples were prepared by peeling them from the quartz substrate using double sided tape (GUITAR was removed all at once, leaving a bare substrate behind), and testing was done on the freshly exposed underside. The HOPG was prepared in a similar manner using double sided tape (HOPG can be readily exfoliated a few layers at a time), and testing was carried out on the freshly exposed underside. Each measurement was carried out at five different spots at each time interval, and the values were averaged.

### 2.8. Electrode Fabrication and Electrochemistry

The GUITAR and N-GUITAR working electrodes were prepared as described in previous study [1]. Freshly exfoliated HOPG electrode was prepared by using adhesive tape, copper tape was used as a current collector and this surface was used within 30 s of cleavage. All electrochemical experiments were performed in an undivided three-electrode cell, with an Ag/AgCl (3 M KCl, aq) reference electrode and a graphite rod as counter electrode. Electrochemical analyses were conducted with either a Bioanalytical Systems CV-50W (West Lafayette, IN, USA) or a Gamry Instruments 1000 (Warminster, PA, USA) potentiostat. All solutions were prepared using house deionized water with further purification through an activated carbon cartridge (Barnstead, model D8922, Dubuque, IA, USA). The cyclic voltammograms were modeled for standard rate constants ($k^0$) using Digisim 3.03b software (Bioanalytical Systems, Inc. West Lafayette, IN, USA). Dopamine oxidation peak current and peak separation values for literature was estimated by using graph grabber v2.0.2.

## 3. Results and Discussion

### 3.1. X-ray Photoelectron Spectroscopy (XPS) Analysis

All XPS peak assignments are based on recent literature [24–27]. The wide scan XPS of N-GUITAR in Figure 2A reveals an atomic percentage of 86.4% C, 9.7% N and 3.9% O based on the 284.1, 398.2 and 532.4 eV binding energy peaks respectively. The based on deconvolved peaks the assignments for C1s (Figure 2B) peaks follow as 68.4% $sp^2$ C (284.6 eV), 21.1% $sp^3$ (286.1 eV), 7.8% C–N and C=N (287.6 eV), 2.7% C=O and COOH (289.4 eV). The material of this investigation is in line with other CVD-grown nitrogen-doped carbon materials that have N atomic content from 1 to 16% [14,28–31]. Also, the $sp^2/sp^3$ atomic ratio of this N-GUITAR is similar to undoped GUITAR in a previous investigation [1]. The O1s peak (Figure 2C) deconvolves to 55.8% N–O and C–O (532.1 eV) and 44.2% N=O and C=O (533.0 3 eV). For the N species in the N1s peaks (Figure 2D) the assignments are 23.7% pyridinic N (398.2 eV), 60.5% graphitic N-center (400.98 eV), 11.8% graphitic N-valley (403.0 eV) and 4% N-oxides (405.3 eV).

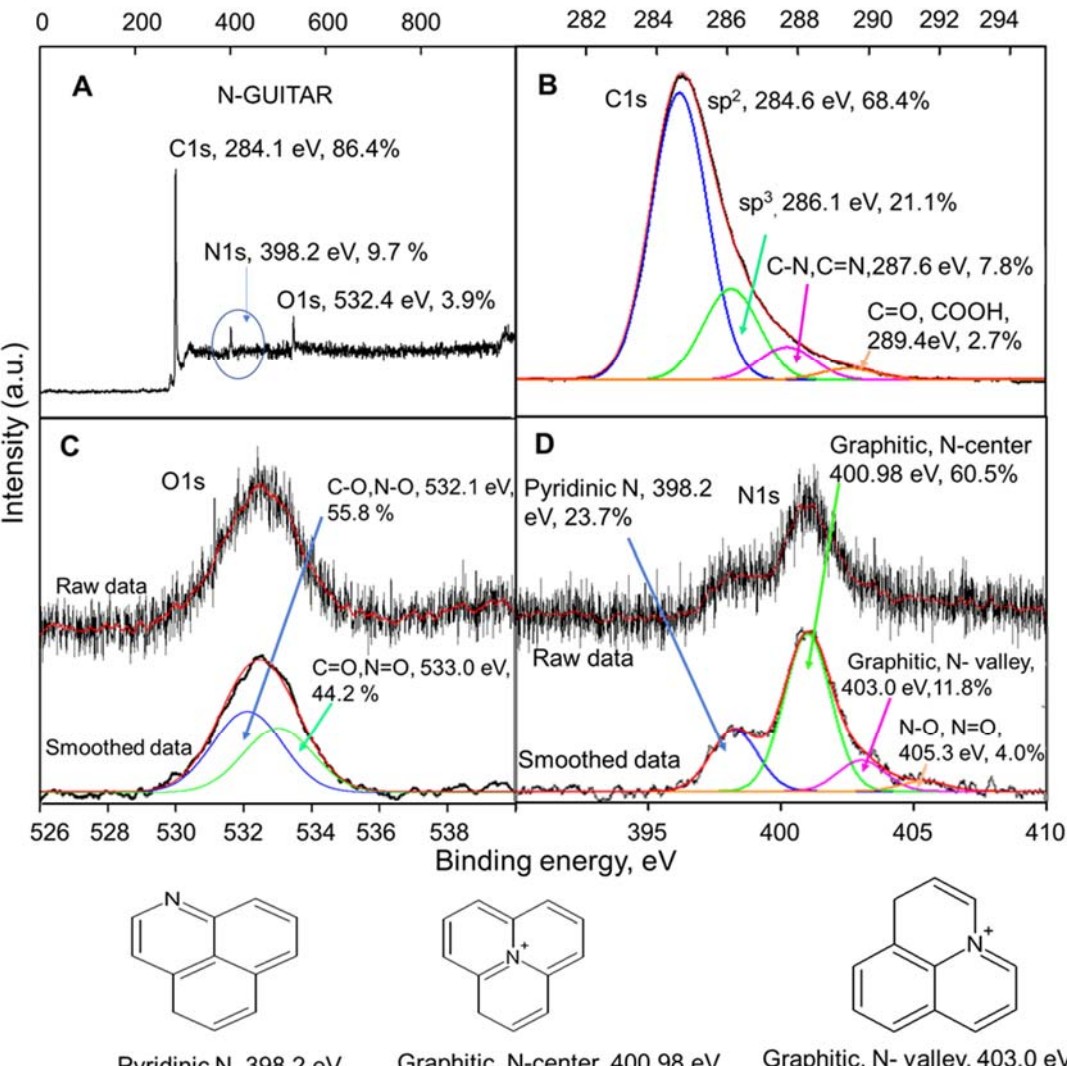

**Figure 2.** XPS analysis of N-doped pseudo graphite with the respective peak assignments and abundances. (**A**) the wide scan spectrum. The C1s (**B**), O1s (**C**) and N1s (**D**) peaks with deconvolved components. The N1s and O1s spectra are presented with the raw and smoothed data using the Savitzky-Golay method. Experimental data is in black and fitted curves are in red. The structures of the nitrogen species are shown at the bottom.

### 3.2. Micrographs

Figure 3 shows the visual and microscopic features of GUITAR and N-GUITAR. The flakes from both forms are undisguisable from each other and resemble high quality graphite. From the scanning electron micrographs (SEM), both a layered basal (BP) and edge plane (EP) morphology are evident. The thickness of the carbon films is 1.3–1.9 μm which increases with deposition time.

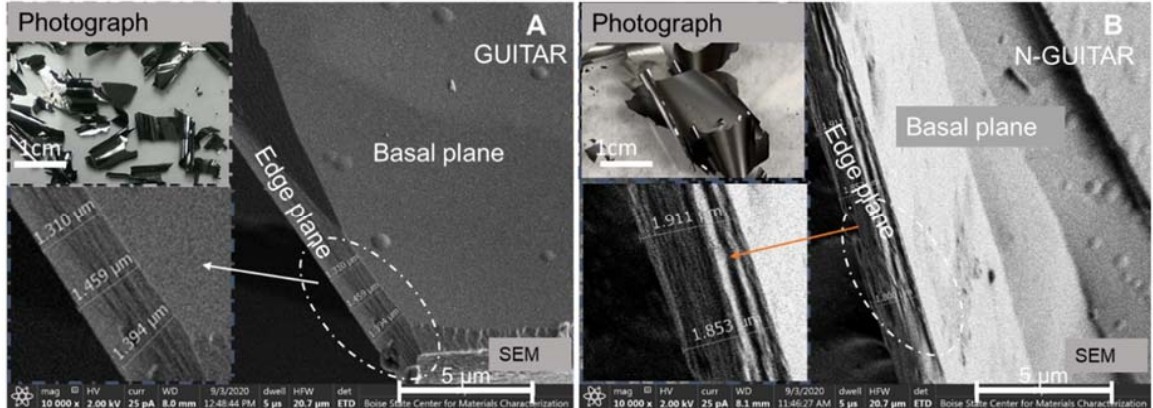

**Figure 3.** Scanning electron micrographs (SEM) of GUITAR (**A**) and N-GUITAR (**B**) with photographs of the flakes in the upper left insets. Both materials exhibit layered morphologies with edge and basal planes. The lower left insets show the SEM's of edge planes in more detail.

### 3.3. Raman Analysis

Raman spectra of N-GUITAR, GUITAR and HOPG are shown in Figure 4. The D-band is related to the structural defects while the G-band describes the degree of graphitization [32]. The HOPG spectrum is used as a reference standard for a low defect material. The GUITAR and N-GUITAR spectra indicate an upshift in the D-band to 1350 from 1340 $cm^{-1}$ and downshift in G- band to 1566 from 1578 $cm^{-1}$ after nitrogen doping. This is consistent with the trends observed in literature [33–35]. The increase in $I_D/I_G$ from 1.15 to 1.66 follows the trends observed in literature which indicates that more defects are introduced with N-doping [33].

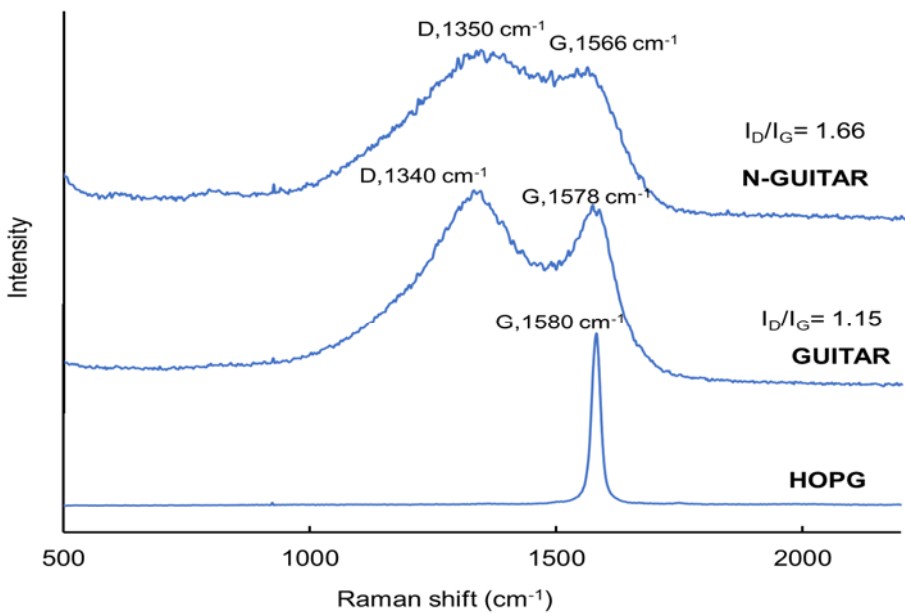

**Figure 4.** Raman spectra for N-GUITAR, GUITAR and HOPG taken with a 532 nm laser.

### 3.4. X-ray Diffraction (XRD) Studies

Figure 5 shows the XRD spectra of N-GUITAR, GUITAR and HOPG. The GUITAR XRD spectra is consistent with previous publications [1]. N-GUITAR shows a basal reflection (002) peak at $2\theta = 26.1°$, which is intermediate of GUITAR and HOPG ($25.4°$ and $26.65°$, respectively). The d-spacing, or average interlayer distance, for N-GUITAR was found to be 341 pm, smaller than that of GUITAR (350 pm) but larger than that of HOPG (334 pm). The d-spacing was calculated from Bragg's law, $n\lambda = 2d\sin(\theta)$, where $n = 1$ and $\lambda$ is the X-ray wavelength (0.154 nm). The average lateral grain size (La) for N-GUITAR and GUITAR were very similar, at 3.3 and 2.9 nm, respectively, making them both nano-crystalline. This is in contrast to HOPG, which will typically range from 30 nm to the low mm range, with the larger grain sizes indicating higher quality [36]. The grain size is calculated from Scherrer's law, $La = K\lambda/\beta\cos(\theta)$, where K is the crystallite shape factor (1.84), $\beta$ is the peak broadening at the half maximum (FWHM), and $\theta$ is the Bragg angle [37]. This indicates that there are no major structural differences between GUITAR and N-GUITAR.

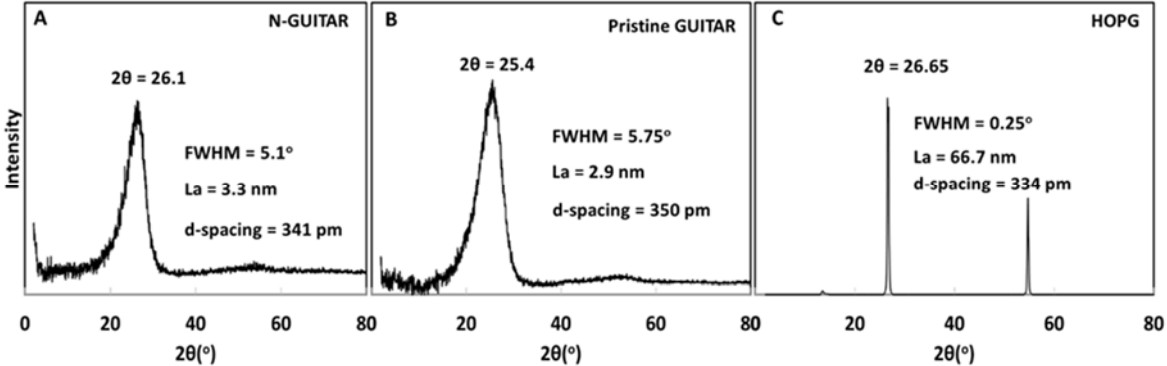

**Figure 5.** Powder XRD spectra of (**A**) N-GUITAR, (**B**) Pristine GUITAR and (**C**) HOPG.

### 3.5. Aqueous Potential Windows

The cyclic voltammograms of GUITAR and N-doped GUITAR in 1M $H_2SO_4$ at 50 mV/s are presented in Figure 6. Based on the cathodic and anodic limits at 200 $\mu A/cm^2$ the potential windows for GUITAR and N-doped GUITAR are 3.0 and 2.7 V, respectively. A decrease in potential window is commonly observed in N-doped materials when compared with their corresponding undoped analogues [38]. This is attributed to faster heterogeneous electron transfer kinetics of the N-doped carbon for hydrogen and oxygen evolution [38,39]. However, both N-GUITAR and GUITAR have aqueous potential windows that are 1 V greater than $sp^2$-hybridized carbon electrodes and match amorphous carbon (a-C) and boron-doped diamond (BDD) materials in this aspect [2,40]. Literature trends suggest that increasing in $sp^3$-C content increases the potential window; however, at cost of lower the HET rate with the $Fe(CN)_6^{4-/3-}$ redox probe [1,41]. The GUITAR electrode counters this trend with a wide potential window and fast HET kinetics [1].

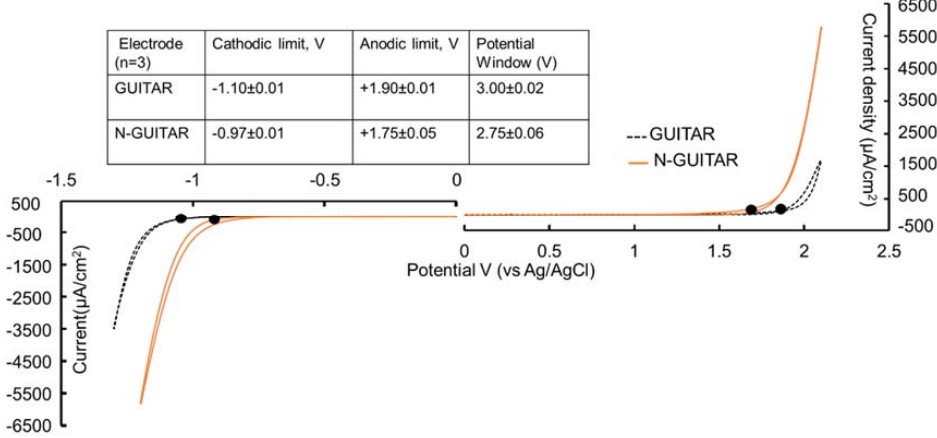

**Figure 6.** Cyclic voltammetric behavior of GUITAR (dashed line) and N-doped GUITAR (solid line) in 1 M $H_2SO_4$ taken at 50 mV/s. The starting potential for both anodic (**right**) and cathodic (**left**) sweeps is 0.0 V. The 200 $\mu A/cm^2$ potential limits are indicted with black dots and the potential windows are listed in the table inset.

### 3.6. Tafel Corrosion Studies

Tafel polarization measurements were carried out in Ar saturated 1.0 M $H_2SO_4$ using a single compartment 1.0 L volume cell under vigorously stirred conditions (see Figure 7). The working electrodes were allowed to equilibrate in electrolyte solutions for ≥ 1 h to attain an equilibrium prior to the start of polarization. The working electrodes were scanned from −0.5 to +0.5 V vs. Ag/AgCl at a scan rate of 1 mV/s. Corrosion current ($i_{corr}$) and corrosion potential ($E_{corr}$) were estimated by extrapolations of the respective Tafel plots [1]. The average $i_{corr}$ of N-GUITAR is $7.3 \times 10^{-9} \pm 3.2 \times 10^{-9}$ $A/cm^2$, a very similar anodic stability to GUITAR ($2.07 \times 10^{-9} \pm 0.7 \times 10^{-9}$ $A/cm^2$). This is three orders of magnitude lower than a graphite rod ($1.82 \times 10^{-6} \pm 1.06 \times 10^{-6}$ $A/cm^2$), which falls into the typical range of other $sp^2$-C electrodes [1,42,43]. It is significant that both N-GUITAR and its undoped counterpart have an anodically stability in the range of a-C and BDD electrodes ($i_{corr}$ typically ranging from of $10^{-9}$ to $10^{-8}$ $A/cm^2$) [44,45]. The corrosion resistance of GUITAR is based on a lack of electrolyte intercalation through interplanar and basal plane defects [2,46,47]. The N-doped form apparently retains this feature. The anodic limit of Figure 6 indicates that there is no intercalation of electrolyte through the BP of either GUITAR or N-GUITAR [2].

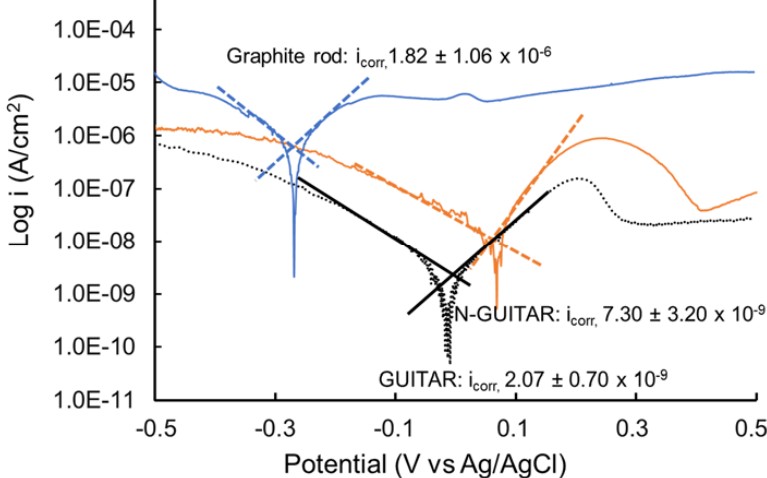

**Figure 7.** Tafel Plots with extrapolated corrosion currents ($i_{corr}$, $A/cm^2$) indicated for each. The average and one standard deviation interval (n = 3) are reported for electrode material.

*3.7. Heterogeneous Electron Transfer (HET) Characteristics of N-GUITAR with Fe(CN)$_6{}^{3-/4-}$*

The Fe(CN)$_6{}^{3-/4-}$ species is a common probe for measuring HET rates across the electrode–electrolyte interface. It is considered an inner-sphere redox couple and is strongly influenced by density of electronic states (DOS) near the Fermi level, adsorbed contaminants and surface chemistry [48–50]. With graphitic materials there is a clear difference in Fe(CN)$_6{}^{3-/4-}$ HET rates between basal (BP) and edge planes (EP). The standard rate constant ($k^0$) for the EP and BP range from $10^{-5}$ to $10^{-1}$ and from $10^{-10}$ to $10^{-6}$ cm s$^{-1}$, respectively [2,51]. Sluggish HET kinetics of graphitic materials at the BP is attributed to low DOS near the Fermi level or from aging in laboratory air, which creates an adsorbed layer of volatile organic compounds [9–11,52]. Figure 8 shows the CV recorded for 1 mM Fe(CN)$_6{}^{3-/4-}$ in 0.1 M KCl at 50 mV/sec. Basal plane GUITAR electrodes are unusual in that the there is a fast calculated $k^0$ of $1.2 \times 10^{-2}$ cm/s and $1.6 \times 10^{-2}$ cm/s for N-GUITAR [1,2]. This lies in the upper range of other N-doped and undoped carbon electrodes ($1.02 \times 10^{-4}$ to $1.5 \times 10^{-2}$ cm/s) [39,40,53,54]. Both materials have fast HET kinetics due to the rich density of structural defects which increase DOS [55].

*3.8. N-GUITAR Is Resistant to Air-Aging*

Several investigators have examined the air-aging effects of HOPG with the Fe (CN)$_6{}^{3-/4-}$ probe [2,11,15,56]. Unwin et al. examined freshly cleaved surfaces of HOPG electrodes that have a CV peak separation (ΔEp) of 58 mV. After 3 h of exposure to air, that surface exhibited a decrease in HET rates with a ΔEp of 450 mV [11]. Liu et al. also conducted a study of HOPG air aging. By storing their electrodes at −15° C they were able to slow the aging effect. The ΔEp increased at a slower rate from 59 mV on a fresh surface to 95 mV at 8 h [15]. Compton and coworkers showed that ΔEp increases from 227 to 596mV at 2 h [56]. A recent body of literature indicates that this effect is from the adsorption of volatile organic compounds (VOCs) onto the surface of HOPG, which creates a barrier to HET. Ellipsometry indicates that this layer is about 0.6 nm thick [15]. Water droplet contact angle (WCA) measurements also support the adsorption hypothesis. Generally, other investigators have found that HOPG and graphene surfaces become more hydrophobic with exposure time, increasing in contact angle from 59–70° to 90–95° [15,16,57,58].

In this investigation a freshly cleaved surface of HOPG gave Fe(CN)$_6{}^{3-/4-}$ ΔEp of 67 mV with a standard deviation of 4 mV over three measurements at 50 mV/s. The ΔEp increases to 165 ± 60 and 221 ± 30 mV at the 1 and 3 h intervals, respectively (Figure 8C). This trend follows literature observations. During that interval the WCA increased from 63.8° ± 5.2° to 80.1° ± 10.8°, again matching literature trends. GUITAR electrodes (Figure 8B) gave increases in ΔEp from 71 ± 2 to 102 ± 5 mV and WCA from 62.6° ± 5.5° to 78.4° ± 7.2° from the 0 to 3 h time interval. These CV trends match that of a previous investigation [2]. Figure 8A demonstrates that N-GUITAR is the most resistant to air-aging effects when compared to literature and to GUITAR. The CV exhibited a minimal increase from 66 ± 5 to 69 ± 3 mV for ΔEp from the fresh surface to 3 h of air exposure. The change in WCA for N-GUITAR again exhibited a minimal increase from 55.8° ± 7.1° to 70.4° ± 5.0° after aging. No other material was observed in the literature that has such resistance to air-aging. This remarkable property indicates excellent prospects for durable electrochemical sensors [4–7,59].

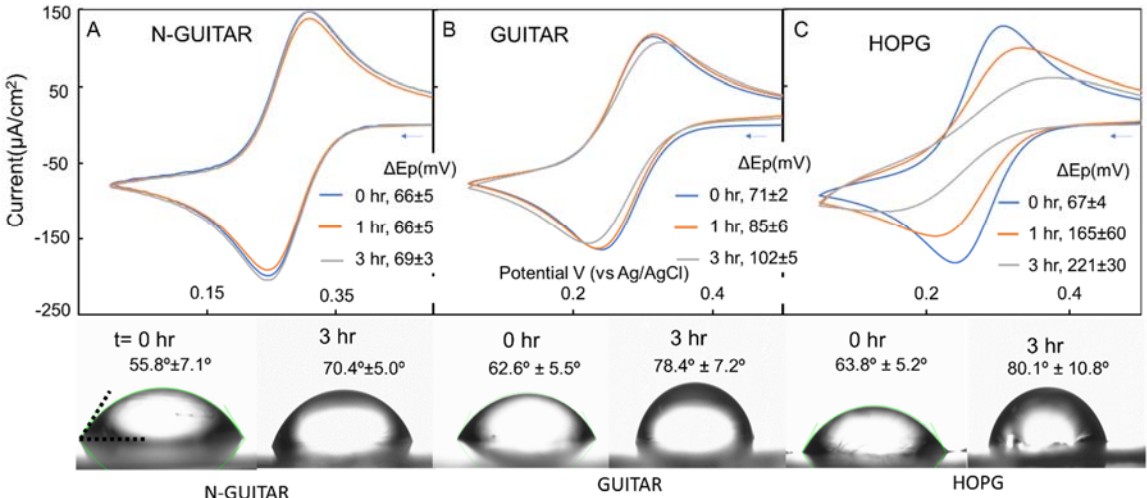

**Figure 8.** Cyclic Voltammetric (CV) studies for the N-GUITAR (**A**), GUITAR (**B**) and HOPG (**C**) electrode with 1 mM Fe (CN)$_6$$^{3-/4-}$ in 0.1 M KCl at 50 mV/sec. CV were recorded in freshly exfoliated surface at 0, 1, and 3 h of air exposure (n = 3). Water contact angle measurement (WCA) on N-GUITAR, GUITAR and HOPG are also shown on their respective positions. WCA measurements were also carried out on freshly exfoliated surfaces at 0, 1, and 3 h of air exposure.

### 3.9. N-Doped GUITAR Is Tolerant to Dopamine Fouling Because of Fast HET Kinetics

Figure 9A shows the CV response of N-GUITAR at relatively high (1 mM) and low concentration (1 μM) of dopamine (DA) in 0.1 M phosphate buffer (pH 7.0). The CV waves are assigned the process seen in step 1 of Figure 10. The ΔEp for 1 mM DA (CV wave seen in Figure 9A) is 83 mV at 100 mV/s. Decreasing ΔEp is associated with faster HET rates [19]. This is the narrowest ΔEp for DA reported in the literature with the exception of EP-pyrolytic graphite, with 59 mV [17]. The ΔEp for GUITAR is 174 mV. Generally, ΔEp varies from 100 to 350 mV for most carbon electrodes [17,19].

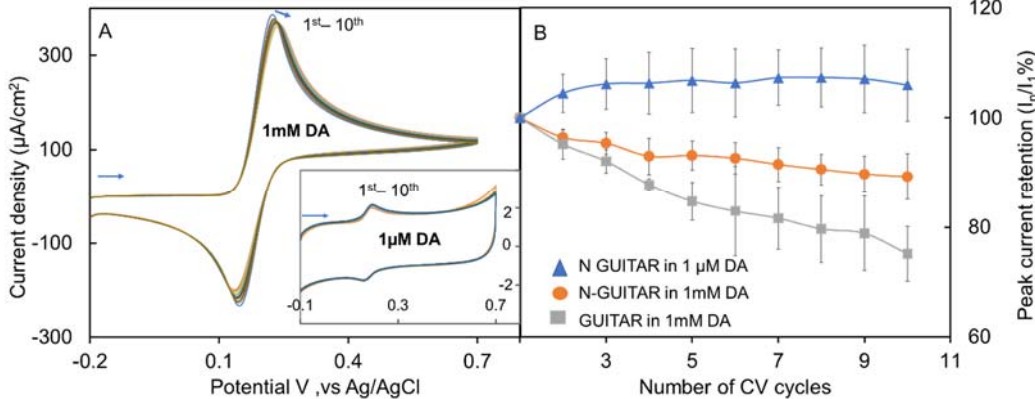

**Figure 9.** (**A**) 10 CV cycles recorded in 1 mM and (inset) 1 μM dopamine in 0.1 M phosphate buffer system (PBS) at pH 7.0 at 100 mV/sec for N-GUITAR electrode. (**B**) Plot of relative oxidative peak current density vs. number of CV runs. The one standard deviation interval for three CV runs are shown with each data point.

Due to their biocompatibility and chemical inertness, carbon electrodes are favored for in vivo voltammetric detection of dopamine [60]. However, they are highly susceptible to fouling from the dopamine polymerization process described in the introduction and the scheme seen in Figure 10. Generally, carbon electrodes tend to lose 20 to 75% of CV current signal after 10 cycles with 1 mM DA [17,19]. Figure 9B plots the peak current with each CV cycle for three trial runs. One standard

deviation error bar is expressed for each point. For N-GUITAR at 1 µM DA there is no loss of peak current over 10 CV cycles. At the higher concentration of 1 mM there is a 10% loss in peak current. The GUITAR electrode experienced a 25% loss in peak current which is in the range of other carbon materials [17,19]. By maintaining the DA peak current, it is apparent that the N-GUITAR surface provides less favorable conditions for the formation of polydopamine.

The reaction scheme for electrode fouling is shown below in Figure 10. The CV waves of Figure 9A A are associated with DA and its oxidized product, o-dopaminoquinone (Step 1). It is the latter that undergoes processes with further electro-oxidation that result in electrode fouling through Steps 2–4 [19,22]. The fraction of o-dopaminoquinone that contributes to fouling can be described by the ratio of CV peak anodic and cathodic currents, $i_{p,a}/i_{p,c}$. A quantity close to one is associated with a lower fraction of o-dopaminoquinone contributing to Steps 2–4. Higher values of $i_{p,a}/i_{p,c}$ indicate that the electrode fouling processes are more facile.

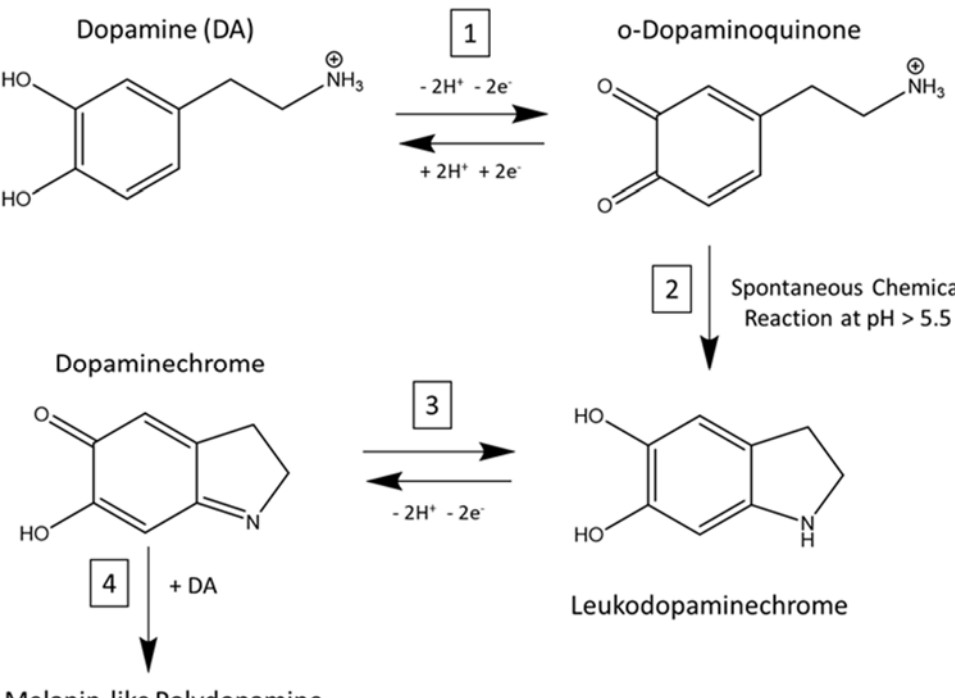

**Figure 10.** Reaction scheme for electrode fouling by electrochemical dopamine oxidation [17,61]. In Step 1, the electrode oxidation and reduction are assigned to the cyclic voltammetric (CV) waves of Figure 9A. Steps 2–4 are the spontaneous series of reactions that result in the formation of the insulating layer of melanin-like polydopamine.

For N-GUITAR electrode, the $i_{p,a}/i_{p,c}$ ratio is 1.3; this is the lowest value reported in literature. For GUITAR, this value is 1.7. This indicates that o-dopaminoquinone is least likely to contribute to fouling using the N-GUITAR material. Figure 11 illustrates the performance of N-GUITAR in comparison with the literature. A plot of $i_{p,a}/i_{p,c}$ vs. ΔEp for DA is shown in Figure 11A, with a clear trend of fast HET rates being associated with a lower $i_{p,a}/i_{p,c}$ ratio. Rapid consumption of o-dopaminoquinone reduces the probability of electrode fouling by Steps 2–4. This effect may be attributed to the HET rate of o-dopaminoquinone reduction back to DA (inverse Step 1). Figure 11B illustrates that low $i_{p,a}/i_{p,c}$ is associated with less $i_{p,a}$ signal loss over 10 CV cycles. Fast HET rates with a variety of redox species on GUITAR electrodes have been discussed in previous publications [2,4,7,62]. This effect is associated with a high DOS arising from structural defects. It is apparent that this effect is enhanced by nitrogen doping GUITAR. The only exception to this trend is the pure sp$^3$ material, BDD

(number 4, Figure 11B), which has relatively slow HET kinetics for DA but exhibited less fouling than is expected from the trend.

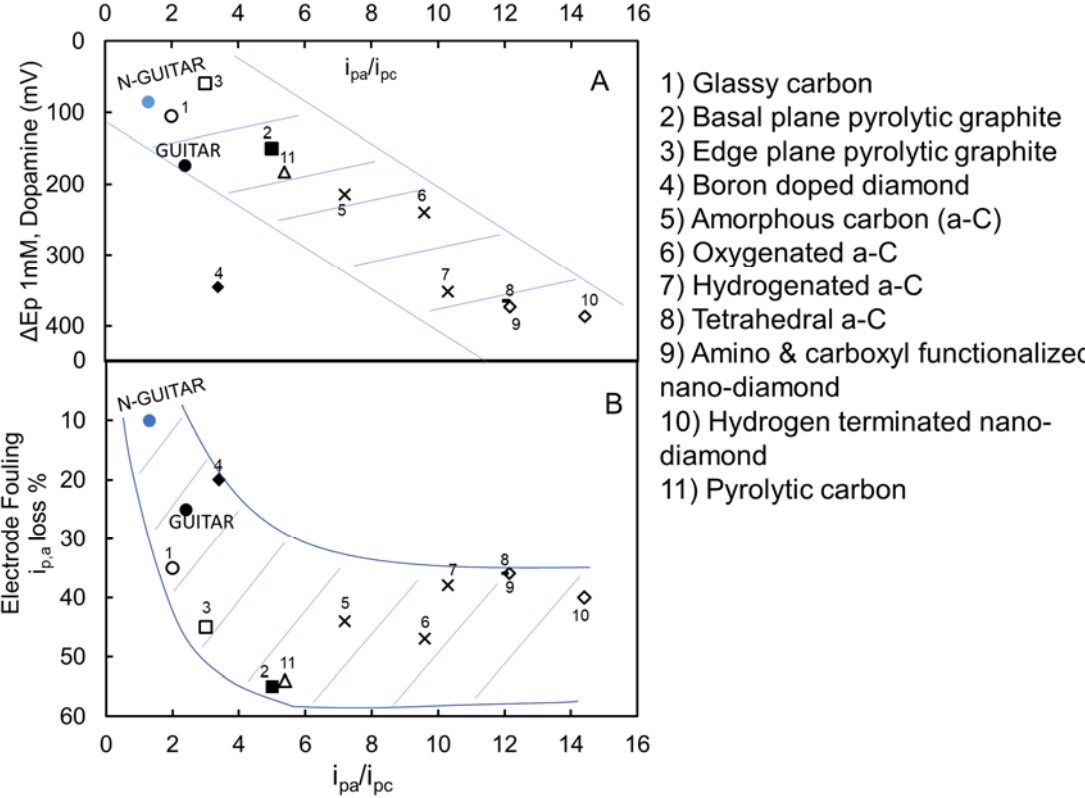

**Figure 11.** The cyclic voltammetric trends regarding HET kinetics of Step 1 in Figure 10. N-GUITAR and GUITAR as well as literature data were recorded in 1 mM DA at pH 7 phosphate buffer solutions [17–19]. The $i_{p,a}/i_{p,c}$ ratio indicates the reversibility of Step 1 and electrode fouling through loss of $i_{p,a}$. (**A**) Plot of $i_{p,a}/i_{p,c}$ vs. $\Delta Ep$ for N-GUITAR, GUITAR and literature carbon. It is clear that a trend forms with fast HET kinetics (narrow $\Delta Ep$) and lower $i_{p,a}/i_{p,c}$. (**B**) Electrode fouling through $i_{p,a}$ loss vs. $i_{p,a}/i_{p,c}$ after 10 cycles. The trends are in the shaded region and show that N-GUITAR has the fastest HET rate (**A**) with the least amount of electrode fouling (**B**).

## 4. Conclusions

The N-GUITAR material maintains the features of GUITAR electrodes, e.g., fast HET rates, wide aqueous potential window and resistance to corrosion. However, it deviates from GUITAR in that it exhibits more resistance to fouling than GUITAR and to other carbon electrodes. The two studies representing the predominant mechanisms of fouling—physical adsorption and polymerization of matrix components—were both observed to be less severe on the surface of N-GUITAR over GUITAR and literature carbon electrodes. In air-aging studies, the WCA measurements indicate that N-GUITAR is less susceptible to VOC physical adsorption than GUITAR or other literature carbon electrodes. In fouling by DA electro-oxidation, N-GUITAR also shows less susceptibility than GUITAR or literature carbons. This is most attributed to fast HET kinetics for the DA/o-dopaminoquinone redox couple on the N-GUITAR electrode. The combination of fast HET and resistance to signal attenuation indicates much promise for N-GUITAR electrodes for DA sensing.

**Author Contributions:** Conceptualization of the project, manuscript writing, and a major portion of the experimental work was done by K.H.; Sample preparation, XRD analysis, manuscript writing, and editing was done by J.M.; SEM micrographs performed by H.Z.; Preparation of GUITAR electrodes was done by F.D.; XPS analysis performed by E.E. and supervised by D.N.M.; Raman analysis done by E.A.; manuscript writing, project administration and supervision done by I.F.C. All authors have read and agreed to the published version of the manuscript.

**Funding:** This research received no external funding.

**Acknowledgments:** The authors acknowledge the university of Idaho for its support of the graduate students through teaching assistantships.

**Conflicts of Interest:** The authors declare no conflict of interest.

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
