# Peer review of "Electrochemical Aspects of a Nitrogen-Doped Pseudo-Graphitic Carbon Material: Resistance to Electrode Fouling by Air-Aging and Dopamine Electro-Oxidation"

_carbon, 2020_

Round 1

Reviewer 1 Report

This manuscript reports N-doped Pseudo-Graphitic Carbon Material (N-GUITAR) for its application in dopamine sensing. N-GUITAR shows promise for a DA sensor with its resistance to electrode fouling by air and fast heterogeneous electron transfer kinetics with Fe(CN)63-/4- with exposure to air. It is suggested for publication at the journal of carbon research with minor revision.

  • The author determined the La of their samples using the Scherrer's equation. However, they choose K as 0.9 instead of 1.84 for La calculation (Phys. Rev. 1941; 59: 693-698). The La values should be recalculated based on the correct K.
  • Figure 2: The experimental data and fitted curves should be clearly labelled. The authors are suggested to color code different XPS peaks. There appeared a few gray lines in the figure, which should be removed.
  • The scale bars are not clear shown in Figure 3. It is suggested to increase the size of the scale bar.

Reviewer 2 Report

The introduction should more clearly explain the novelty claimed in the paper, as several papers from the same team (ref 1 to 7) are proposed but not really summarised or discussed to show the new impact.

The work is nice but a complete comparison with other carbon based materials would enhance the results (like amorphous carbon, BDD, N doped aerogels, N doped CNT...) - except fig 11 where such a comparison is done but not enough explained in the text.

We can also have a question about the reproducibility of the N-GUITAR process (and therefore how many similar samples were used for the experimental results presented ?).

Applications with such a material should be given.

Minor spell errors need to be checked.
